# Microwave-Assisted Ionic Liquid-Catalyzed Selective Monoesterification of Alkylphosphonic Acids—An Experimental and a Theoretical Study

**DOI:** 10.3390/molecules26175303

**Published:** 2021-08-31

**Authors:** Nikoletta Harsági, Réka Henyecz, Péter Ábrányi-Balogh, László Drahos, György Keglevich

**Affiliations:** 1Department of Organic Chemistry and Technology, Budapest University of Technology and Economics, 1521 Budapest, Hungary; harsagi.nikoletta@vbk.bme.hu (N.H.); reka422@gmail.com (R.H.); 2Medicinal Chemistry Research Group, Research Centre for Natural Sciences, 1117 Budapest, Hungary; abranyi-balogh.peter@ttk.mta.hu; 3MS Proteomics Research Group, Research Centre for Natural Sciences, 1117 Budapest, Hungary; drahos.laszlo@ttk.hu

**Keywords:** alkylphosphonic acid, monoesterification, selectivity, microwave, ionic liquid, mechanism, energetics, theoretical calculations

## Abstract

It is well-known that the P-acids including phosphonic acids resist undergoing direct esterification. However, it was found that a series of alkylphoshonic acids could be involved in monoesterification with C_2_–C_4_ alcohols under microwave (MW) irradiation in the presence of [bmim][BF_4_] as an additive. The selectivity amounted to 80–98%, while the isolated yields fell in the range of 61–79%. The method developed is a green method for P-acid esterification. DFT calculations at the M062X/6–311+G (d,p) level of theory (performed considering the solvent effect of the corresponding alcohol) explored the three-step mechanism, and justified a higher enthalpy of activation (160.6–194.1 kJ·mol^−1^) that may be overcome only by MW irradiation. The major role of the [bmim][BF_4_] additive is to increase the absorption of MW energy. The specific chemical role of the [BF_4_] anion of the ionic liquid in an alternative mechanism was also raised by the computations.

## 1. Introduction

Dialkyl phosphonates and their derivatives may be important intermediates or starting materials in different reactions [1], they have importance in the pharmaceutical industry [2] and in biochemistry [3]. Phosphonates find application also in environmental chemistry [4] and as flame retardants [5]. The typical preparation of phosphonates (**III**) involves the reaction of aryl- or alkylphosphonic dichlorides (**I**) with alcohols or phenols, or the Arbuzov reaction of trialkyl phosphites (**IV**) with alkyl- or aryl halides (Scheme 1) [1,2].

According to a newer approach, the syntheses were based on the direct esterification of the P(O)OH moiety of the P-acids. Among a series of phosphinic acids, phenyl-*H*-phosphinic acid (**V**) was also subjected to direct esterification under microwave (MW) conditions using [bmim][PF_6_] ionic liquid (IL) as the catalyst [6]. In the absence of an IL, the esterification was not so efficient [7]. The alkyl phenyl-*H*-phosphinates (**VI**) so obtained were oxidized by *m*-chloro-perbenzoic acid to the corresponding phosphonic ester-acids (**VII**) (Scheme 2) [7]. The MW-assisted direct esterification of ester-acids **VII** furnished the dialkyl phenylphosphonates (**VIII**) in lower yields of 25–62% [7]. A more appropriate protocol was, when phenylphosphonic acid (**IX**) was converted to the mono ester **VII** under MW irradiation and IL catalysis, and the ester-acid (**VII**) so obtained was converted to the diester (**VIII**) by alkylation (Scheme 2) [8].

The ionic liquids comprising a cation and an anion are regarded “green” solvents due to their inflammability, negligible vapor pressure (low volatility), and solvation power [9,10,11]. Ionic liquids are especially good solvents for metal complexes [12]. Although the primary function of ionic liquids is to serve as solvents, they find more and more applications as catalysts/additives [13].

In this article, we describe the MW-assisted monoesterification of alkylphosphonic acids. Beyond the preparative work, we also aimed at the investigation of the theoretical background of the target reaction.

## 2. Results and Discussion

### 2.1. Preparative Experiments on the Microwave-Assited Monoesterification of Alkylphosphonic Acids

The model reaction studied experimentally and theoretically is shown in Scheme 3. The monoesterification of the alkylphosphonic acids (**1A**–**D**) was performed using 15 equivalents of the alcohol and 10% of [bmim][BF_4_] under MW conditions. [Bmim][BF_4_] was selected on the basis of our earlier studies [8]. Formation of some of the diester (**3A–D**) was inevitable. The pure monoesters **2A**–**D/a**–**d** could be obtained by converting them to the corresponding sodium salts by treatment with 10% aqueous NaOH, removing the diester (**3A**–**D/a**–**d**) by extraction with dichloromethane, and liberating the free acid by acidification with hydrochloric acid.

The results obtained with methylphosphonic acid (**1A**) are summarized in Table 1. Due to the volatility and taking into account the pressure limit of the MW reactor, the esterification with EtOH had to be carried out at 160 °C. The almost complete conversion was attained after an irradiation of 5 h. The ratio of the mono and the diester (**2Aa** and **3Aa**) was 94–2 (Table 1/Entry 1). The reactivity of PrOH and *^i^*PrOH was significantly different, after a reaction at 180 °C for 3 h, the conversion was 97% and 77%, respectively, while the proportion of the mono and diesters (**2Ab**/**2Ac** and **3Ab**/**3Ac**) was 78–19 and 74–3, respectively (Table 1/Entries 2 and 3). Using *^i^*PrOH, the conversion was complete after 5 h (Table 1/Entry 4). Employing BuOH, the appropriate reaction conditions were either 180 °C/3 h or 200 °C/1.25 h. In these cases, the ratio of **2Ad** and **3Ad** was ca. 81:19 (Table 1/Entries 5 and 6). On the effect of prolonged heating, the ratio of the diester **3Ad** increased to 55% (Table 1/Entry 7). It is noteworthy that a control experiment carried out in the lack of [bmim][BF_4_] took place in a conversion of 19% (Table 1/Entry 6, footnote “b”). Hence, the role of IL is unambiguous. In another comparative experiment performed on conventional heating without IL, the conversion was 10% (Table 1/Entry 6, footnote “c”). This proves that both MW and IL are needed for efficient esterification.

The results of the direct esterification of ethylphosphonic acid **1B** can be found in Table 2. Using EtOH at 160 °C, completion required > 6 h. After an irradiation of 6 h, the conversion was 95%, and the ratio of the mono and the diester (**2Ba** and **3Ba**) was 87:8 (Table 2/Entry 1). The esterification with PrOH and *^i^*PrOH at 180 °C was almost complete after 3.5 h and 7 h, respectively, leading to ester mixtures (**2Bb**–**3Bb** and **2Bc**–**3Bc**) 79:16 and 91:7, respectively (Table 2/Entries 2 and 3). Using BuOH the completion took >4 h at 180 °C. After an irradiation of 3.5 h, the ratio of the mono and the diester (**2Bd** and **3Bd**) was 89:7 (Table 2/Entry 4). Performing the esterification at a somewhat higher temperature of 200 °C, the ratio of **2Bd** and **3Bd** was 79:20 (Table 2/Entry 5).

The results obtained during the esterification of propylphosphonic acid (**1C**) are listed in Table 3. One can see that the data on the reaction times, conversions, and monoester—diester ratios are rather similar to those obtained for the esterification of ethylphosphonic acid (**1B**). This means that the reactivity of the ethyl- and propylphosphonic acid is rather similar in the monoesterification under discussion.

As can be seen from Table 4, butylphosphonic acid (**1D**) was the less reactive in the series. The esterification with EtOH at 165 °C was not even complete after 7 h (Table 4/Entry 1). Applying PrOH and *^i^*PrOH at 200 °C, the reaction time was 4 h, and, as extrapolated, 10 h, respectively (Table 4/Entries 3 and 4). With BuOH, the esterification required 200 °C/3 h.

The ester-acid species **2A**–**D/a**–**d** were isolated from the best mixtures by treatment with NaOH/H_2_O, extraction with dichloromethane, and liberating the ester-acid (**2A**–**D/a**–**d**) with hydrochloric acid. A final extraction with dichloromethane furnished the pure monoesters that were identified by ^31^P NMR chemical shifts and HRMS (See Experimental, Table 7). From among the 16 ester-acids, 9 were described earlier, and were identified by us by ^31^P NMR shifts [14,15,16,17]. The remaining 7 ester-acids (**2Bb**, **2Bd**, **2Cb**, **2Da**, **2Db**, **2Dc**, and **2Dd**) were also characterized by us by ^1^H and ^13^C NMR spectral data (see Appendix A). The minor components, dialkyl alkylphosphonates **3A**–**D/a**–**d** were identified by ^31^P NMR and HRMS (See Experimental, Table 8).

The synthetic method developed for the selective monoesterification of phosphonic acids (**1**) is green, as it avoids the application of P-chlorides.

### 2.2. Theoretical Calculations on the Direct Esterification of Alkylphosphonic Acids

We have analyzed the energetics of the direct esterification of phosphonic acids (**1**, R = Me, Et, Bu) with alcohols (MeOH, BuOH) using DFT computations at the M062X/6–311+G (d,p) level of theory considering the solvent effect of the corresponding alcohol (Scheme 4, Table 5). Based on our previous model [18], we proposed a reaction complex containing three alcohol reagents and two phosphonic acids, where one alcohol molecule acts as the reagent in the monoesterification of one of the phosphonic acid units. The other P- and ROH species in the reaction complex are responsible for the proton transfer chain that promotes the formation of the new P-O bond and the departure of an H_2_O molecule. The formation of the reaction complex (**4**) is highly exothermic marked by a Δ*H* of (−123)–(−146.6) kJ·mol^−1^, but the significant decrease in entropy (Δ*S*) by the assessment of five molecules results in an increased Gibbs free energy value (Δ*G* = 46.0–66.0 kJ·mol^−1^). The first transition state (**TS1**) belongs to the addition of the reacting alcohol onto the P atom of the P=O function resulting in intermediate **5**. This reaction is moderately endothermic with an activation enthalpy requirement of 70.7–94.5 kJ·mol^−1^. The second step is the removal of a hydroxy group from species **5** that is realized via dehydration taking a proton from a nearby OH unit. Notably, the second phosphonic acid unit remains intact and acts only as a proton transfer additive. Interestingly, although the intermediate following **TS1** has a higher enthalpy, a similar or slightly lower Gibbs free energy value may suggest a slight stabilization after the transition. The enthalpy of activation requirement for **TS2** belonging to the dehydration is, as compared to the starting level [19], 160.6–194.1 kJ·mol^−1^. The Gibbs free energy values are also over 150 kJ·mol^−1^. The higher barrier may be overcome at a higher temperature of 180–200 °C utilizing MW assistance [20,21]. Finally, the product complex (**6**) breaks up, and the liberation of the product (**2**) is slightly exothermic as compared to the enthalpy level of the starting reactants (meaning an enthalpy gain of 8.4–15.8 kJ·mol^−1^).

The question is raised, what the role of the IL may be? Well, it is assumed that similar to earlier cases, the role of [bmim][BF_4_] is to increase the MW absorbing ability of the medium [22,23].

It is worth noting that according to the energetics, the esterification of phenylphosphonic acid that was studied earlier [8] goes with similar enthalpy, energy, and entropy changes (see the last row of Table 5) as the alkylphosphonic acids (**1**), meaning that the reactivity of the phosphonic acids in monoesterification is not influenced much by the nature of the substituent.

The enthalpy diagram for the monoesterification of ethylphosphonic acid (**1B**) with butanol is shown in Figure 1. The gross enthalpy of activation is 160.6 kJ·mol^−1^.

Next, the [bmim][BF_4_]-promoted esterification was studied for a few alkylphosphonic acid–alcohol combinations (Scheme 5, Table 6). Similar to the additive-free way, the reaction goes through a multicomponent complex (**7**) containing the phosphonic acid, the BF_4_^−^ anion, and two alcohol molecules. The formation of this complex (**7**) is exothermic regarding enthalpy [the gain is (−56.8)–(−63.4) kJ·mol^−1^], but considering the Gibbs free energy values, again an increase may be observed. The esterification, in this case, is a single-step process involving a rate-determining transition state (**TS3**) with a somewhat lower Δ*H* value (162.4–171.0 kJ·mol^−1^) as compared to the other pathway shown in Scheme 4. This leads to a product complex (**8**) that results in the formation of the monoesterified product (**2**) after decomplexation.

The enthalpy diagram for the [bmim][BF_4_]-promoted monoesterification of ethylphosphonic acid (**1B**) with butanol is shown in Figure 2. The enthalpy of activation is 171.0 kJ·mol^−1^ that is comparable with that of the gross value (160.6 kJ·mol^−1^) of the other mechanism (Figure 1).

Analyzing the datasets computed for the direct esterification and BF_4_-promoted version, one might conclude that according to the calculations, at least at this level, there is no significant discrimination among the P-substituents or among the alcohols. However, when comparing the two mechanistic pathways, one can see that while the direct way (*A*) is a two-step process, the ionic-liquid-promoted version (*B*) involves just one step. Another difference is that the formation of the primary reaction complexes (**4** in “*A*” and **7** in “*B*”) needs the association of five and four molecules, respectively. So, route *B* seems to be simpler. However, the decisive factor may be that the enthalpy of activation values are comparable for the two kinds of mechanisms (*A* and *B*): for the selected examples represented in Figure 1 and Figure 2, the enthalpy of activation value is 160.6 kJ·mol^−1^ and 171.0 kJ·mol^−1^, respectively. These high values may be overcome by MWs. The final message is that in the cases studied, both mechanisms (*A* and *B*) may be operative. The role of ionic liquid may be merely to increase MW absorption (as in earlier cases [20,21]), but it is also possible that the BF_4_ anion of the ionic liquid participates chemically, and hence promotes the esterification.

In summary, an MW-assisted, IL-promoted method was elaborated for the selective monoesterification of alkylphosphonic acids with simple alcohols. This is a green method, as avoids the use of P-chlorides. The mechanism explored by high-level theoretical calculations suggested the formation of a ring associate comprising two phosphonic acid molecules and three alcohol units, the nucleophilic attack of the alcohol on the P atom of the P=O moiety, and dehydration exhibiting a gross enthalpy of activation of 160.6–194.1 kJ·mol^−1^. The high barrier could be overcome by the beneficial effect of MWs. A [bmim][BF_4_] additive ensured the efficient absorption of MWs. An alternative mechanism involving the participation of the BF_4_ anion of the ionic liquid was also substantiated, but the activation energy requirement of this option was also high (162.4–171.0 kJ·mol^−1^).

## 3. Experimental

### 3.1. General

The ^31^P, ^13^C, and ^1^H-NMR spectra were taken on a Bruker DRX-500 spectrometer operating at 202.4, 125.7, and 500 MHz, respectively. The couplings are given in Hz. LC-MS measurements were performed with an Agilent 1200 liquid chromatography system coupled with a 6130 quadrupole mass spectrometer equipped with an ESI ion source (Agilent Technologies, Palo Alto, CA, USA).

### 3.2. Use of the ^31^P NMR Spectra in Quantitative Analysis

The composition of the reaction mixture was determined by the integration of the areas under the corresponding peaks of the starting material and product in the ^31^P NMR spectra.

### 3.3. General Procedure for the Direct Esterification of Akylphosphonic Acids in the Presence of Ionic Liquids

A mixture of 1.44 mmol of alkylphosphonic acid (methylphosphonic acid: 0.14 g, ethylphosphonic acid: 0.16 g, propylphosphonic acid: 0.18 g, butylphosphonic acid: 0.20 g), 21.72 mmol of alcohol (1.28 mL of ethanol, 1.62 mL of propanol, 1.68 mL of *i*-propanol, 2.0 mL of butanol) and 0.144 mmol (27 µL) [bmim][BF_4_] was stirred under MW conditions (max 100 W). After evaporation and flash column chromatography (silica gel, DCM–MeOH 97:3), the reaction mixture was analyzed by ^31^P NMR spectroscopy. The crude products of the best experiments were purified by extraction. The residue obtained after evaporation was taken up in 5 mL of CH_2_Cl_2_, and the solution was stirred with a mixture of 0.60 mL of 10% NaOH/H_2_O. The phases were separated, the water phase was acidified with 0.14 mL of 37% hydrochloric acid, and stirred with 5 mL of CH_2_Cl_2_. The organic phase was dried (Na_2_SO_4_) and its concentration afforded the corresponding ester-acids (**2A**–**D**) as colorless oils.

Identification data of ester acids **2A**–**D/a**–**d** and diesters **3A**–**D/a**–**d** are listed in Table 7 and Table 8, respectively.

### 3.4. Additional Spectral Data for the New Ester-Acids

#### 3.4.1. Monopropyl Ethylphosphonate (**2Bb**)

^13^C NMR (CDCl_3_) Δ: 6.3 (d, ^2^*J*_P,C_ = 6.7, PCH_2_CH_3_), 10.0 (s, OCH_2_CH_2_CH_3_), 19.0 (d, ^1^*J*_P,C_ = 145.1, PCH_2_), 23.7 (d, ^3^*J*_P,C_ = 6.1, OCH_2_CH_2_), 66.5 (d, ^2^*J*_P,C_ = 6.9, OCH_2_); ^1^H NMR (CDCl_3_) Δ: 0.96 (t, *J* = 7.4, 3H, CH_3_), 1.14–1.21 (m, 3H, CH_3_), 1.66–1.80 (m, 4H, PCH_2_, CH_2_), 3.98 (q, *J* = 6.8, 2H, OCH_2_), 9.62 (s, 1H, OH).

#### 3.4.2. Monobutyl Ethylphosphonate (**2Bd**)

^13^C NMR (CDCl_3_) Δ: 6.4 (d, ^2^*J*_P,C_ = 6.7, PCH_2_CH_3_), 13.6 (s, OCH_2_CH_2_CH_2_CH_3_), 18.7 (s, OCH_2_CH_2_CH_2_), 19.0 (d, ^1^*J*_P,C_ = 145.0, PCH_2_), 32.5 (d, ^3^*J*_P,C_ = 6.0, OCH_2_CH_2_), 64.7 (d, ^2^*J*_P,C_ = 6.8, OCH_2_); ^1^H NMR (CDCl_3_) Δ: 0.94 (t, *J* = 7.3, 3H, CH_3_), 1.13–1.21 (m, 3H, CH_3_), 1.37–1.45 (m, 2H, CH_2_), 1.63–1.79 (m, 4H, PCH_2_, CH_2_), 4.02 (q, *J* = 6.7, 2H, OCH_2_), 8.32 (s, 1H, OH).

#### 3.4.3. Monopropyl Propylphosphonate (**2Cb**)

^13^C NMR (CDCl_3_) Δ: 10.0 (s, OCH_2_CH_2_CH_3_), 15.2 (d, ^3^*J*_P,C_ = 17.3, PCH_2_CH_2_CH_3_), 16.0 (d, ^2^*J*_P,C_ = 4.8, PCH_2_CH_2_), 23.8 (d, ^3^*J*_P,C_ = 6.0, OCH_2_CH_2_), 27.9 (d, ^1^*J*_P,C_ = 142.7, PCH_2_), 66.4 (d, ^2^*J*_P,C_ = 6.8, OCH_2_); ^1^H NMR (CDCl_3_) Δ: 0.96 (t, *J* = 7.3, 3H, CH_3_), 1.03 (t, *J* = 7.1, 3H, CH_3_), 1.64–1.77 (m, 6H, PCH_2_, 2 CH_2_), 3.97 (q, *J* = 6.6, 2H, OCH_2_), 7.96 (s, 1H, OH).

#### 3.4.4. Monoethyl Butylphosphonate (**2Da**)

^13^C NMR (CDCl_3_) Δ: 13.5 (s, PCH_2_CH_2_CH_2_CH_3_) 16.3 (s, OCH_2_CH_3_), 23.6 (d, ^2^*J*_P,C_ = 16.8, PCH_2_CH_2_CH_2_), 24.3 (s, PCH_2_CH_2_), 25.6 (d, ^1^*J*_P,C_ = 143.0, PCH_2_), 61.0 (d, ^2^*J*_P,C_ = 5.1, OCH_2_); ^1^H NMR (CDCl_3_) Δ: 0.92 (t, *J* = 7.3, 3H, CH_3_), 1.33 (t, *J* = 6.7, 3H, CH_3_), 1.38–1.45 (m, 2H, CH_2_), 1.57–1.64 (m, 2H, CH_2_), 1.71–1.78 (m, 2H, PCH_2_), 4.06–4.12 (m, *J* = 7.3, 2H, OCH_2_), 10.08 (s, 1H, OH).

#### 3.4.5. Monopropyl Butylphosphonate (**2Db**)

^13^C NMR (CDCl_3_) Δ: 10.0 (s, OCH_2_CH_2_CH_3_), 13.6 (s, PCH_2_CH_2_CH_2_CH_3_) 23.6 (d, ^3^*J*_P,C_ = 17.6, PCH_2_CH_2_CH_2_), 23.8 (d, ^2^*J*_P,C_ = 5.9, OCH_2_CH_2_), 24.2 (d, ^3^*J*_P,C_ = 4.8, PCH_2_CH_2_), 25.5 (d, ^1^*J*_P,C_ = 143.8, PCH_2_), 66.4 (d, ^2^*J*_P,C_ = 6.8, OCH_2_); ^1^H NMR (CDCl_3_) Δ: 0.92 (t, *J* = 7.3, 3H, CH_3_), 0.96 (t, *J* = 7.4, 3H, CH_3_), 1.38–1.45 (m, 2H, CH_2_), 1.56–1.64 (m, 2H, CH_2_), 1.66–1.78 (m, 4H, PCH_2_, CH_2_), 3.97 (q, *J* = 6.8, 2H, OCH_2_), 9.62 (s, 1H, OH).

#### 3.4.6. Monoisopropyl Butylphosphonate (**2Dc**)

^13^C NMR (CDCl_3_) Δ: 13.6 (s, PCH_2_CH_2_CH_2_CH_3_) 23.7 (d, ^3^*J*_P,C_ = 17.4, PCH_2_CH_2_CH_2_), 24.0 (d, ^2^*J*_P,C_ = 3.1, OCH(CH_3_)_2_), 24.4 (d, ^3^*J*_P,C_ = 2.9, PCH_2_CH_2_), 26.1 (d, ^1^*J*_P,C_ = 143.8, PCH_2_), 70.0 (d, ^2^*J*_P,C_ = 6.0, OCH_2_); ^1^H NMR (CDCl_3_) Δ: 0.91 (t, *J* = 7.3, 3H, CH_3_), 1.33 (d, *J* = 6.0, 6H, 2 CH_3_), 1.37–1.45 (m, 2H, CH_2_), 1.56–1.62 (m, 2H, CH_2_), 1.69–1.76 (m, 2H, PCH_2_), 4.65–4.72 (m, *J* = 6.5, 1H, OCH), 10.46 (s, 1H, OH).

#### 3.4.7. Monobutyl Butylphosphonate (**2Dd**)

^13^C NMR (CDCl_3_) Δ: 13.6 (s, OCH_2_CH_2_CH_2_CH_3_), 13.6 (s, PCH_2_CH_2_CH_2_CH_3_), 18.8 (s, OCH_2_CH_2_CH_2_) 23.7 (d, ^3^*J*_P,C_ = 17.5, PCH_2_CH_2_CH_2_), 24.3 (d, ^2^*J*_P,C_ = 4.5, PCH_2_CH_2_), 25.6 (d, ^1^*J*_P,C_ = 143.2, PCH_2_), 32.5 (d, ^3^*J*_P,C_ = 5.9, OCH_2_CH_2_), 64.7 (d, ^2^*J*_P,C_ = 6.7, OCH_2_); ^1^H NMR (CDCl_3_) Δ: 0.93 (dt, *J* = 9.9, 7.3, 6H, 2 CH_3_), 1.37–1.45 (m, 4H, 2 CH_2_) 1.57–1.77 (m, 6H, PCH_2_, 2 CH_2_), 4.01 (q, *J* = 6.7, 2H, OCH_2_), 9.73 (s, 1H, OH).

### 3.5. Theoretical Calculations

DFT computations at the M062X/6–311+G (d,p) level of theory were performed considering the solvent effect of the corresponding alcohol using the SMD solvent model with the Gaussian 09 program package [27,28,29]. The geometries of the molecules were optimized in all cases, and frequency calculations were also performed to assure that the structures are in a local minimum or in a saddle point. The conformations of the reported structures have been determined by conformational analysis. The solution-phase Gibbs free energies were obtained by frequency calculations as well. The G values obtained were given under standard conditions, the corrected total energies of the molecules were taken into account. Entropic and thermal corrections were evaluated for isolated molecules using standard rigid rotor harmonic oscillator approximations. That is, the Gibbs free energy was taken as the “sum of electronic and thermal free energies” printed in a Gaussian 09 vibrational frequency calculation. Standard state correction was taken into account. The transition states were optimized with the QST3 or the TS (Berny) method. Transition states were identified by having one imaginary frequency in the Hessian matrix, and IRC calculations were performed in order to prove that the transition states connect two corresponding minima.

## Data Availability

Not applicable.

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
