# Peer review of "Microwave-Assisted Ionic Liquid-Catalyzed Selective Monoesterification of Alkylphosphonic Acids—An Experimental and a Theoretical Study"

_molecules, 2021, doi:10.3390/molecules26175303_

Round 1

Reviewer 1 Report

The work presented describes the access to monoesterified alkylphosphonic acids exploiting the effect of a selected ionic liquid and microwaves. This way the use of rather toxic P-chlorides is avoided.

The proposed approach, although an extension of previous results by the same authors on different substrates, is of value and deserves to be published after the following changes.

The introduction section should be implemented by adding relevant literature on ionic liquids to define the context for non-experts.

General on ILs : https://doi.org/10.1007/s12551-018-0419-2, https://doi.org/10.1016/j.molliq.2019.112038.

electric conductivity: https://doi.org/10.1016/j.molliq.2017.03.036

negligible vapor pressure and low volatility: https://doi.org/10.1039/C6CP01948J,

inflammability or Low flammability: https://doi.org/10.1016/j.jlp.2020.104196.

strong thermal stability: 10.1021/acssuschemeng.0c02473.

surface activity: 10.1016/j.molliq.2021.115988

solvation power of organic compounds: https://doi.org/10.1039/C9NJ03239H,

solvation of inorganic compounds: 10.1039/C9RA04042K

and some application:

Reaction solvents https://doi.org/10.1515/pac-2016-1008

Reaction catalyst: 10.1016/j.mcat.2020.110854

Biochemistry: https://doi.org/10.1016/j.ssi.2017.11.012

electro- chemistry: https://doi.org/10.1016/j.coelec.2018.03.005,

analytical chemistry https://doi.org/10.1021/acs.analchem.8b04710

pharmaceutics and medicine: 10.3390/ijms21218298,

Furthermore, some examples of the use of ionic liquids in synthesis as solvents and/or catalysts and the effect of microwaves on reaction outcomes should be commented.

Line 16: is the word chemoselective correct in the monoesterification reaction?

Line 73: applying does not sound correct (employing?)

Table 5 and Table 6 should be moved to ESI.

Line 219: please specify the operating conditions for 1H and 13C NMR

Line 287: please add the heading for the DFT part

Author Response

Dear Referee 1:

"The work presented describes the access to monoesterified alkylphosphonic acids exploiting the effect of a selected ionic liquid and microwaves. This way the use of rather toxic P-chlorides is avoided.

The proposed approach, although an extension of previous results by the same authors on different substrates, is of value and deserves to be published after the following changes."

Thanks for the positive attitude.

"The introduction section should be implemented by adding relevant literature on ionic liquids to define the context for non-experts."

In our work, we applied ILs not as solvents, but (only) as a catalyst, so an intensive discussion of ILs, in our opinion,  would not be justified. However, Referee 1 is completely right that something should be said on ILs. For this, we incorporated generale literature, their potential as solvents (for also metal complexes) and as catalysts/additives. Five new references  (9-13) were incloded, four from the list of the referee.

"Furthermore, some examples of the use of ionic liquids in synthesis as solvents and/or catalysts and the effect of microwaves on reaction outcomes should be commented."

As described above, we did not apply ILs as solvents. The application of ILs as catalysts was mentioned (also) in a general sense supported by new reference 13 (covering a recent review of ours in this topic).

ILs may act as MW absorbers, but this was mentioned in the old version. See references 22 and 23 in the middle of p. 6.

"Line 16: is the word chemoselective correct in the monoesterification reaction?"

OK, it is indeed a borderline case. We corrected, and wrote only "selectivity".

"Line 73: applying does not sound correct (employing?)"

"applying" was replaced by "employing".

"Table 5 and Table 6 should be moved to ESI."

In other MDPI publications, the list of energetics was tolerated, as the numbers are expressive. Pls allow to keep the two tables in the body of the ms.

"Line 219: please specify the operating conditions for 1H and 13C NMR"

Was specified.

"Line 287: please add the heading for the DFT part"

Was added.

Thanks agin for the the constructive remarks,  G. Keglevich + Niki

Reviewer 2 Report

The paper is about microwave-assisted ionic liquid-catalyzed monoesterification of alkylphosphonic acids. This is a very interesting and well-written work. I can certainly recommend it for publication in Molecules. In my opinion, only minor corrections are needed.

Suggestions:

  1. Please check the multiplicity of the signals on the 1H NMR spectrum (about 4.ppm for all compounds, and 4.68 for 2Dc) – they do not agree with the structure of the compounds (it might be better to describe them as multiplet).
  2. Compound 2Da, 13C NMR – 16.3 ppm – it should be OCH2CH3
  3. Table 6 and 7 – I suggest adding columns: “[M + Na]+calculated” and “formula” (then there will be three columns HRMS) - it will be more clear
  4. Table 7, 3Ac, 3Bc and 3Cc – there is no HRMS (181, 195, 209)
  5. 15 – it should be Org. Biomol. Chem. 2012, 10, 2011-2018.
  6. Supporting Information - I propose to describe the spectra (solvent, frequency, etc.)

Questions:

  1. What about the reaction with methanol? You described reactions with ethanol, propanol, and butanol, but not with methanol – why?
  2. At which frequency the NMR spectra were recorded - see 3.1 General, lines 219-220?
  3. Compound characterization: what about IR spectroscopy and the physical state of the compounds?

Author Response

Dear Referee 2:

"The paper is about microwave-assisted ionic liquid-catalyzed monoesterification of alkylphosphonic acids. This is a very interesting and well-written work. I can certainly recommend it for publication in Molecules. In my opinion, only minor corrections are needed."

Thanks for the compliments.

"Suggestions:

  1. Please check the multiplicity of the signals on the 1H NMR spectrum (about 4.ppm for all compounds, and 4.68 for 2Dc) – they do not agree with the structure of the compounds (it might be better to describe them as multiplet)."

OK, you were absolutely right that this issue needs some re-consideration. In case of the molecules incorporating the P(O)-O-CH2 moiety, the protons appear as a doublet of triplets. Well, this is transformed to a virtual quartett. This was checked. However, in case of the ethyl and isopropyl compounds, this cannot be the case, for this, indeed a multiplet is justified. This was corrected.

"Compound 2Da, 13C NMR – 16.3 ppm – it should be OCH2CH3"

This was corrected.

"Table 6 and 7 – I suggest adding columns: “[M + Na]+calculated” and “formula” (then there will be three columns HRMS) - it will be more clear"

This was corrected.

"Table 7, 3Ac, 3Bc and 3Cc – there is no HRMS (181, 195, 209)"

Our mass spectroscopic colleague (Dr L. Drahos) in a few cases could not utilize the peaks belonging to the very minor products due to their low intensities. Pls disregard from HRMS in a few cases, as only minor products are involved.

"15 – it should be Org. Biomol. Chem. 2012, 10, 2011-2018."

The name of the journal was italized.

"Supporting Information - I propose to describe the spectra (solvent, frequency, etc.)"

The information was completed.

"Questions:

What about the reaction with methanol? You described reactions with ethanol, propanol, and butanol, but not with methanol – why?"

MeOH is too volatile. We could no°t ensure 180-200 °C due to the pressure limit of 20 bar. For this MeOH was not tested.

"At which frequency the NMR spectra were recorded - see 3.1 General, lines 219-220?"

This info was inserted.

"Compound characterization: what about IR spectroscopy and the physical state of the compounds?"

We did not obtain IR data. The 31P, 1H and 13C NMR, along with HRMS seemed to be satisfactory. Registration  of the IR data would require a series of re-synthesis and than obtaining the IR data, that would take more than a month. Can you disregard from this request?

Thanks again for the constructive remarks,

György Keglevich + Niki

Round 2

Reviewer 1 Report

The Authors improved the manuscript satisfactorily. Therefore, I recommend publication of the work as is.